# Modeling Driver Behavior in Road Traffic Simulation

**DOI:** 10.3390/s22249801

**Published:** 2022-12-14

**Authors:** Teodora Mecheva, Radoslav Furnadzhiev, Nikolay Kakanakov

**Affiliations:** Department of Computer Systems and Technologies, Technical University Sofia, Plovdiv Branch, 4017 Plovdiv, Bulgaria

**Keywords:** Intelligent Transportation Systems, Smart City, SUMO

## Abstract

Driver behavior models are an important part of road traffic simulation modeling. They encompass characteristics such as mood, fatigue, and response to distracting conditions. The relationships between external factors and the way drivers perform tasks can also be represented in models. This article proposes a methodology for establishing parameters of driver behavior models. The methodology is based on road traffic data and determines the car-following model and routing algorithm and their parameters that best describe driving habits. Sequential and parallel implementation of the methodology through the urban mobility simulator SUMO and Python are proposed. Four car-following models and three routing algorithms and their parameters are investigated. The results of the performed simulations prove the applicability of the methodology. Based on more than 7000 simulations performed, it is concluded that in future experiments of the traffic in Plovdiv it is appropriate to use a *Contraction Hierarchies* routing algorithm with the default routing step and the *Krauss* car-following model with the default configuration parameters.

## 1. Introduction

Simulation is the main approach in presenting systems for which it is difficult and expensive to perform real experiments. The degree of correspondence between the real system and the simulation describes the validity of the model. The acceptability of the degree of correspondence depends on the purpose of the further experiments [1].

Road traffic is one of the most intensive domains of simulation modeling due to the great interest in transport and the many pressing issues in this field. Road traffic simulation models are usually agent-based. This makes them suitable for dynamic enrichment, integration of complex behaviors, and modeling intelligent human behavior. The drawback is that the complexity of the model complicates maintenance [1,2].

Road traffic simulation models include many components: the road network, traffic load data, the choice of simulation tool (traffic simulator), the model of interaction between agents, and parameter settings. Model evaluation and calibration is often complex and goal-specific. The use of real input data increases the correspondence to the real world but complicates the validation process [3,4].

Fitting the input data to the simulator (traffic demand estimation) is one of the important tasks to solve. Traffic demand estimation converts input traffic information (lifestyle statistics and detector data) into single-trip routes using routing algorithms [5,6].

Another important component of the road traffic simulation model is the driver behavior model. This takes into account characteristics such as mood, fatigue, and responses to distracting conditions [7,8]. The relationships between external factors and the way drivers perform tasks, such as choices of headway during car-following, gap acceptance, overtaking, speed choice, and lateral and longitudinal control, can also be represented in the model [9,10,11].

Table 1 presents studies reflecting different aspects of driver behavior.

In articles [7,12,13], the importance of advanced real-time driver information systems is emphasized. The authors of [7] provided the results of a field experiment that confirms the advances of the presented application: increased driver attention and pursuit of safe driving. The authors of [12] prove the effectiveness of the proposed V2V-based information system via simulation. The conclusions of [13] are ambiguous. The results of a two-stage survey and field experiment show that dynamic route recommendations do not always yield results.

Articles [9,10] investigate the effect of distraction during driving. While [9] uses a traffic simulator to analyze the experimental data, [10] compares two data sets: with and without distraction influences. The authors of [9] reported a deterioration in traffic flow when texting and talking on the phone, while [10] concluded that looking sideways for 2 s did not affect drivers’ reactions.

The authors of [14,15,16] examine the factors of driver behavior that influence traffic accidents in different cultures. Based on the three studies, it can be concluded that the factors that affect traffic safety vary widely among different cultures.

Articles [8,11,17] describe the application of machine learning techniques for driver behavior data. All applied methods—Kalman filter, linear regression, logistic regression, gradient boosting, and random forest—are suitable for analyzing driver behavior.

The current article proposes a methodology for choosing simulation model components that best reflect driver habits. The methodology is not tied to a specific simulation tool.

In Section 2, the description of the methodology is presented. In Section 3, an implementation of the methodology is proposed using SUMO road traffic simulator and video input data. Section 4 describes parallel implementation of the methodology. Section 5 presents experimental results using four car-following models and three routing algorithms with different parameters and input data sets.

## 2. Methodology Description

The proposed methodology aims to select the routing algorithm and car-following model and their settings that best reflect driving habits.

A car-following model is a function of position, speed, and acceleration. The most commonly used are micro models, as, with their many adjacent parameters, a higher degree of correspondence to the real system can be achieved. Examples of popular microscopic car-following models are Gipps, Krauss, and Widermann [4,5].

Routing algorithms are the main part of traffic demand estimation. They formalize networks as weighted graphs and compute routes. Examples of commonly used routing algorithms are Prim, Kruskal, and Dijkstra [18,19]. Routing algorithms take as input data the number of vehicles or the ratio between traffic flows in given locations. In cases where data are set for many locations, the routing algorithms may experience certain difficulties, and as a result, simulated values deviate from the specified values.

The proposed methodology selects a routing algorithm and a car-following model and their parameters that most closely correspond to driver habits by choosing the combination with the smallest deviation between the input and simulated values.

The simulation setup encompasses the road map, traffic light cyclograms, and traffic loads for road lines. Each vector of the Cartesian product of the possible values of the car-following model and routing algorithm and their parameters are input into the simulator. The discrepancy between the simulated data and the real load data shows how well the respective configuration matches driver behavior. Figure 1 depicts the scheme of the methodology.

For example, the configuration (Krauss, minGap = 1 m, tau = 2 m, Dijkstra, step = 10 s) is an input vector for a simulation. This means that the simulation uses a Krauss car-following model with a minimum distance when stopped of −1 m, a minimum distance when moving of −2 m, and a Dijkstra routing algorithm with a route recalculation step of −10 s. After simulation execution, the information about the number of cars that passed through each lane is compared with the input data. The discrepancy between these two quantities in percent is calculated for each line. The average value of the discrepancies for the certain simulation is a metric for the validity of the configuration. the configuration with the lowest discrepancy is considered to best match driving habits.

## 3. Experimental Setup

The part of the road network in the city of Plovdiv between “Ruski” Boulevard (to the east), “Osvobozhdenie” Boulevard (to the west), “Gladstone” Street and “Princess Maria Luisa” Boulevard (to the north), and “Hristo Botev” Boulevard and Saint Petersburg Boulevard (to the south) is integrated in SUMO (Simulator for Urban Mobility) (Figure 2).

SUMO is an open-source multimodal traffic simulator created by the German Aerospace Center [20]. SUMO offers variety of micro models (Krauss, Widermann, Wagner, Kerner, Intelligent Driver Model, and new models and modifications) and a meso model. Time representation is continuous. The traffic demand input also has variety of options: randomization, OD matrices, classical four-step approach, flow definitions, ratios, and using data from observation points. The output data for analysis can be aggregated data about traffic: the number of vehicles, speed, delays at certain points, emissions, floating car data, and traffic light configuration [21,22].

The data for the traffic loads in the experiment are obtained through virtual detectors built on the basis of road cameras. They are extracted from the database of the Municipality of Plovdiv in the form of Excel reports. Each report contains information about the number of vehicles that entered the line of a single intersection per hour. The available reports cover the periods from 15 January to 27 January 2021. Reports for the period 19 February–19 March 2021 for three of the junctions are added to other reports, and an additional set of simulations is performed with the extended data set.

As illustrated in [23], a repeating pattern is observed on working days and on holidays. That is why the experimental part is divided in two: working days and holidays. Each data set is grouped by hour. An average value of each group is an input for a single road line of the simulation.

The actual duration of the traffic light cycles is used based on the municipality’s documentation.

For the purposes of the investigation, three routing algorithms and four car-following models and their configuration parameters are selected (Table 2, Table 3 and Table 4). All three routing algorithms that are available in SUMO are investigated. All car-following models that have stable implementation and no more than six parameters are investigated. The set of studied parameters is chosen so that the default values should belong to the set and are not extreme.

Several Python scripts are implemented to prepare the simulation inputs, run the simulations, and process the output data.

The ControllerReports.py script converts the municipality reports into two CSV (comma-separated values) input files—working day (wCFG.csv) and holiday (hCFG.csv)—by calculating average values per hour for 24 h.

CallFlowrouter.py calculates set of routes and traffic flows from given detectors in the network. The script uses the SUMO tool flowrouter by setting different measurement intervals (values 10, 20, 30, 40, and 50 min are set).

TrafficDemand.py performs dynamic user assignment and runs the SUMO simulation. For dynamic user assignment, the SUMO tool duarouter is used. At this stage, the routing algorithm and car-following model and all their parameters are set. The result is a Dua.xml file that contains all the traffic information for the SUMO simulation. The output of each simulation and the configuration files are saved in a separate folder named after the configuration input.

CalculateDiscr.py compares the number of vehicles that passed each road line from the simulation to the input data in the corresponding CSV input file and calculates the average discrepancy for each simulation.

## 4. Parallel Implementation

The proposed methodology requires evaluation of many different simulation configurations: the Cartesian product of the possible values of the car-following model and routing algorithm and their parameters. All configurations have no data dependencies (neither in input nor in output values). Some authors [24] show that the use of computing resources varies depending on the configuration. They measure the CPU, memory, and disk usage of SUMO and show that for the same road network, the usage may vary depending on the model. A resource-aware workflow for automated scheduling running multiple simulations simultaneously can provide many benefits, especially in reducing time to produce all results needed.

TraficSim implements the control logic for executing a single simulation. It generates the routes and configuration files, executes the simulation, and saves all outputs. This application with all of its dependencies is packaged inside Docker image. To automate and parallelize the execution of simulations with different input configurations, we used an Apache Airflow WMS instance running on a Kubernetes cluster inside a private cloud environment. Airflow is an open-source platform that can programmatically build, schedule, and monitor workflows. They are defined in a Python-based descriptor called a DAG (directed acyclic graph). Inside a DAG, the parameters, execution order, and level of parallelism of each group of simulations are configured as separate tasks. Airflow then uses the Kubernetes API to schedule a pod with the Docker image and configuration for each task in the DAG. Kubernetes scheduling is resource-aware and actively manages workloads. For each scheduled simulation, we set a “Resource request” based on the average of a couple of test runs, and “Resource limits” that restrict the maximum that can be used by a single simulation. All configuration files and the results of the simulation are saved to a cloud object storage bucket.

## 5. Results

A total of 7539 simulations were run; among them, 7103 were with simulation step of 1, and 36 were with smaller simulation steps. The big disadvantage of the more-precise simulation steps is that it requires much more computational resources and/or time [25].

### 5.1. Working Day Data Set Simulations

All possible configurations for *Krauss* (450) and *Wagner* (450) car-following models over the working day data set are simulated. The *Wiedemann* and *Modified Krauss* car-following models are performed for, respectively, 828 and 825 simulations over the working day data set. Additional sets of simulations with different simulation steps and extended input data are performed in order to investigate the factors that influence discrepancy (Table 5).

The minimum mean discrepancy between simulated and real values in the working day data set is 21.598% for *Krauss* and 21.599% for *Wagner*. In both car-following models, the minimum discrepancy is observed in all simulations with routing algorithm *Contraction Hierarchies*, the *routing step*, *minGap*, and *tau* do not affect the results. All simulations of *Modified Krauss* and *Wiedemann* car-following models over the working day data set are configured with the *Dijkstra* routing algorithm and show the same discrepancy −22.824%. The discrepancy between simulated and real values is the same with different values of *minGap* and *tau*.

An additional set of 368 simulation of the *Modified Krauss* car-following model over the working day data set are performed. They are all configured with the *Contraction Hierarchies* routing algorithm and constant values of *minGap* and *tau*. The discrepancies between simulated and real values are the same with different values of *acceleration*, *deceleration*, *emergency deceleration*, and *sigma* at −21.598%.

An additional 30 simulations are performed with simulation steps of 0.01 over the working day data set with the *Contraction Hierarchies* routing algorithm with steps of 50 s and the *Krauss* car-following model. All simulations show the same discrepancy of −32.700%.

An additional four simulations are performed with a simulation step of 0.001 over the working day data set with the *Contraction Hierarchies* routing algorithm with steps of 50 s and with the *Krauss* car-following model. The minimum discrepancy is 30.454%, and the deviation is 0.932%.

An additional two simulations are performed with a simulation step of 0.001 over the working day data set, the *Dijkstra* routing algorithm with steps of 10 s, the *Wagner* car-following model with *minGap = 1*, and *tau*, respectively, of 0.5 or 0.25. The minimum discrepancy in the two simulations is 27.11%, and the deviation is 4.32%.

An additional 90 simulations are performed over the extended working day data set with routing steps of 50 s and with the *Krauss* car-following model. The minimum discrepancy is 24.941%. This is observed with all simulations with a routing algorithm with *Contraction Hierarchies*.

An additional 90 simulations are performed over the extended working day data set with routing steps of 50 s and with the *Wagner* car-following model. The minimum discrepancy is 24.941%. This is observed with all simulations with a routing algorithm with *Contraction Hierarchies*.

A total of 3137 simulations over the working day data set are performed. The minimum discrepancy is 21.598%. This is observed in smaller data set with the *Contraction Hierarchies* routing algorithm and *Modified Krauss* and with the *Krauss* car-following models. The simulations over the extended working day data set show higher discrepancy.

### 5.2. Holiday Data Set Simulations

All possible configurations for *Krauss* (450) and *Wagner* (450) car-following models over the holidays data set are simulated. For the *Wiedemann* and *Modified Krauss* car-following models, we performed, respectively, 828 and 825 simulations over the holiday data set. Additional sets of simulations with different simulation steps, extended input data, and fixed parameters are performed in order to investigate the factors that influence discrepancy (Table 6).

The minimum mean discrepancy between simulated and real values in the holiday data set is 46.990% with the *Krauss* model and 46.776% with the *Wagner* model. In both car-following models, the minimum discrepancy is observed in all simulations with the with the *Dijkstra* routing algorithm. The *routing step*, *minGap*, and *tau* do not affect the results.

All 1412 simulations of the *Modified Krauss* car-following model over the holiday data set are configured with the *Dijkstra* routing algorithm and return a discrepancy of 46.776%. The discrepancy between simulated and real values is the same with different configuration values.

All 1409 simulations of with the *Wiedemann* car-following model over the holiday data set are configured with with the *Dijkstra* routing algorithm. The minimum discrepancy is 46.776%. The discrepancy between simulated and real values is the same with different configuration values.

An additional set of 500 simulation with the *Modified Krauss* car-following model over the holiday data set are performed. They are all configured with the *Dijkstra* routing algorithm and constant values of *minGap* and *tau*. The discrepancy between simulated and real values is the same with different values of *acceleration*, *deceleration*, *emergency deceleration*, and *sigma* at −46.776%.

An additional 90 simulations are performed over the extended holiday data set with a routing step of 50 s and the *Krauss* car-following model. The minimum discrepancy is 22.649%. This is observed in all simulations with the routing algorithm *Contraction Hierarchies*.

An additional 90 simulations are performed over the extended holiday data set with a routing step of 50 s and the *Wagner* car-following model. The minimum discrepancy is 22.649%. This is observed with all simulations with the routing algorithm *Contraction Hierarchies*.

A total of 4402 simulations over the holiday data set are performed. The minimum discrepancy is 22.649%. This is observed with the extended holiday data set, the *Contraction Hierarchies* routing algorithm, and the *Krauss* and *Wagner* car-following models.

## 6. Discussion

A methodology for determining the components of road traffic simulations to describe driving habits is proposed. The methodology defines a car-following model and a routing algorithm and their parameters based on road detectors. Two implementations of the methodology via Python and SUMO are proposed: parallel and sequential. The results of the performed simulations prove the applicability of the methodology.

Currently, results of more then 7000 simulations are available, including configurations with four car-following models over the holiday and working day data sets in short and extended variants. The majority of the performed simulations are configured with the default routing step due to limitations in computational resources. The step of the routing algorithm and the additional car-following model parameters do not affect the discrepancy with simulation steps of 1 and 0.01. With simulation step 0.001, the configuration parameters affect discrepancy.

On working days, driver behavior is best described by the *Contraction Hierarchies* routing algorithm and the *Krauss* and *Modified Krauss* car-following models.The *Wagner* car-following model also shows good results. The smallest discrepancy is observed with the small data set (15–27 January 2021). When extending the data with reports for the period 19–21 February 2021, the discrepancy increases.

The probable reason for the worse performance of the routing algorithm with the extended data set is that the driver behavior pattern changes over time, and mixing data from different periods may lead to data contamination.

In the original holiday data set (15–27 January 2021), the *Dijkstra* routing algorithm and the *Modified Krauss*, *Wagner*, and *Wiederman* car-following models show the best performance. The minimum discrepancy in the holiday data set is observed in the extended data set with the *Contraction Hierarchies* routing algorithm and the *Krauss* and *Wagner* car-following models.

In the original holiday data set, there are only a few records. When adding the data for 19–21 February 2021 to the holiday set, discrepancy decreases significantly. The probable reason for this result is that in larger data sets, some of the measurement errors affect the averages less. The different performance of the routing algorithm on the holiday data set confirms the effectiveness of the *Contraction Hierarchies* routing algorithm under heavy loads.

The possible reason for the different results with the original and expanded data set is that driver behavior varies at different periods and depends on many preconditions. Another possible reason for the different results when expanding the data set is random measurement errors.

It can be concluded that the *Contraction Hierarchies* routing algorithm is suitable for heavy loads, while *Dijkstra* is suitable for lower loads or when data are missing. When using the default simulation step, the routing step and additional model parameters do not affect the final result. The results regarding the car-following model are the most contradictory. In the extended and shortened data sets, different models show minimal discrepancy. For working days, the *Krauss* and *Modified Krauss* models show the best results, while for holidays, *Krauss* and *Wagner* are the best.

## 7. Future Work

The described methodology suggests fitting the simulation parameters to real traffic data regarding the number of vehicles that pass on an hourly basis in order to reach a more realistic simulation model. This idea can be developed using more detailed data about traffic in the city: for example, the types of vehicles and their sizes, the purpose of the trip, the presence of public events, etc.

The described methodology offers to find the most suitable car-following model and routing algorithms and their parameters. It is possible to extend the methodology by adding more components to be tuned: for example, line-change model and its parameters.

Another option for extending the methodology is to look for simulation model components and parameters on an hourly basis instead of working days and holidays.

The proposed implementation via SUMO and Python is a good starting point for future research. The methodology realization with the same input data and different traffic simulators would help to evaluate the universality of the proposed approach. Repeating the experiments with the current implementation and data from other cities or different periods would help the completeness of the research.

The performed experiments may be a good starting point for setting up the SUMO road traffic simulator for future traffic studies in the city of Plovdiv. Based on the preformed experiments, it can be concluded that in future work it is appropriate to use the *Contraction Hierarchies* routing algorithm with the default routing step and the *Krauss* car-following model with the default configuration parameters.

Examining the selected road network with different sets of input data and a simulation step less than 0.001 and the exhaustion of all possible 484,200 configurations would be indicative of the components of the simulation model that best describe the behavior of drivers in the city of Plovdiv. This would require a massive amount of time or computing power. One of the possible future directions of work is finding heuristics and developing a feedback-based workflow.

## Figures and Tables

**Figure 1 sensors-22-09801-f001:**
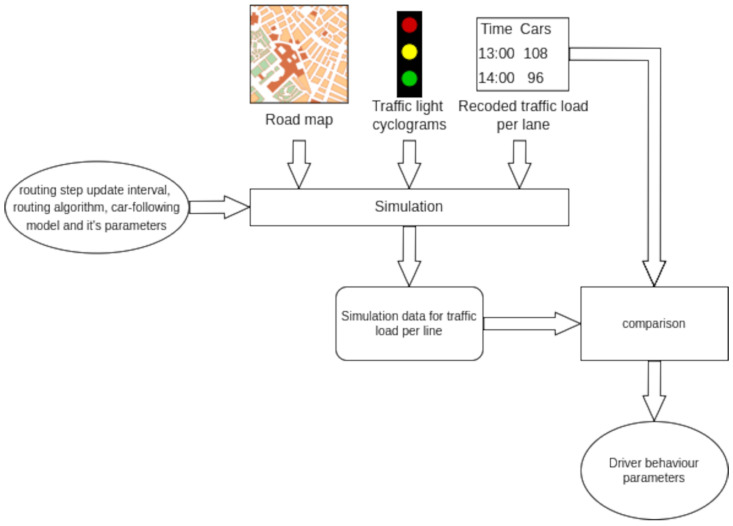
Methodology diagram.

**Figure 2 sensors-22-09801-f002:**
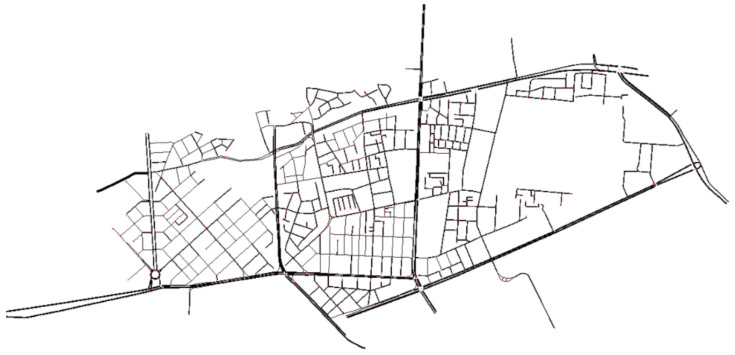
Map being used.

**Table 1 sensors-22-09801-t001:** Different aspects of driver behavior.

Title	Objective	Type of Analyzed Data	Conclusion
Driver Rating: A mobile application to evaluate driver behavior [7]	To test experimentally how sending feedback to drivers in real time affects the manner of driving	Vehicle sensor and smartphone data	The field experiment confirms the effectiveness of the Driver Rating mobile application.
“Machine learning methods for driver behaviour classification” [8]	To investigate techniques for driver behavior detection and evaluation	Raw and processed sensor data and video recordings of trips	The applicability of machine learning classification methods for driver behavior evaluation is confirmed. Three types of driving are classified: normal, drowsy, and aggressive.
“Car following and microscopic traffic simulation under distracted driving” [9]	To investigate car-following models in the context of distracted activities	Data extracted from driving simulator	Simulation experiments over TRANSMODELER traffic simulator with General Motors and Intelligent Driver Model car-following models show deterioration of traffic flow when texting and to some extent when talking on the phone.
“The influence of attention distraction on the drivers’ behaviour” [10]	To examine how time spent focusing attention on roadside advertisements affects safety and driving performance	Data extracted from driving simulator	Investigation of speed, accelerator pedal pressure intensity, and steering wheel angle indicates that taking eyes off the road for 2 s does not significantly affect driver distraction.
“Drivers’ behaviour on expressways: headway and speed relationships” [11]	To study drivers’ car-following behaviour on Malaysian high-speed highways	Hourly traffic data collected via Automatic Traffic Counter connected to pneumatic tubes	Real data are processed via linear regression. The results show that driver behavior is influenced by the types of highway facilities.
“An extended car-following model considering the drivers’ characteristics under a V2V communication environment” [12]	To increase safety and comfort during driving	Traffic simulation	The conducted experiment proved that vehicle-to-vehicle communication can improve traffic stability, safety, and fuel economy.
“Empirical study of effect of dynamic travel time information on driver route choice behavior” [13]	To evaluate the effect of information on driver behavior depending on driver age and historical data.	Pre-run questionnaire, sensor data from field experiment, and post-run questionnaire.	When drivers know the routes well or have more experience, real-time information affects them less. Older drivers are less likely to take risks.
“Assessing the road traffic crashes among novice female drivers in Saudi Arabia” [14]	To evaluate factors that affect road accidents caused by novice female drivers	Questionnaire	Age is not a significant influencing factor. Female novice drivers who are single, divorced/widowed, employed, and have higher individual incomes are at higher risk of getting into car accidents.
“Application of the AHP-BWM Model for Evaluating Driver Behavior Factors Related to Road Safety: A Case Study for Budapest” [15]	To dissect and rank the significant driver behavior factors related to road safety in Budapest	Questionnaire	Driver behavior factors are classified in a three-level hierarchical structure: “Aggressive violations”, “fail to apply brakes in road hazards”, “drive with alcohol use”, and “disobey traffic lights” are distinguished as most significant.
“Analyzing the Importance of driver behavior criteria related to road safety for different driving cultures” [16]	To examine the significant driver behavior criteria in different cultures	Questionnaire	Each country has its own traffic safety issues related to driver behavior.
“Real Time Estimation of Drivers’ Behaviour ” [17]	To estimate the Intelligent Driver Model parameters	Real traffic data	The factors that affect vehicle motion characteristics are: driver lane-change behavior, the number of vehicles in the opposite lane, vehicle type in the opposite lane, and shoulder width.
“Modeling driver behavior in road traffic simulation”	To present a methodology for driver behavior modeling in traffic simulation	Real traffic data	-

**Table 2 sensors-22-09801-t002:** Routing algorithms.

Routing Algorithm	Description
Dijkstra	The simplest and slowest
Astar	Uses a metric for bounding travel time to direct the search and is often faster than Dijkstra
Contraction Hierarchies	Very efficient when a large number of queries is expected

**Table 3 sensors-22-09801-t003:** Car-following models.

Model	Notes	Examined Parameters
Modified Krauss	The Krauss-model with some modifications. The default model used in SUMO. There are 6 rather than the usual 2 tuning parameters.	minGap, accel, decel, emergencyDecel, sigma, tau
Krauss	The original Krauss-model.	minGap, tau
Wagner	A model by Peter Wagner using Todosiev’s action points.	minGap, tau
Wiedemann	Still under development. Some tuning parameters are hard-coded into the model.	minGap, tau, security, estimation

**Table 4 sensors-22-09801-t004:** Model parameters.

Parameter	Notes	Default Value	Examined Values	Unit
minGap	Minimum gap when standing	2.5	1, 1.5, 2, 2.5, 3	m
accel	The acceleration ability of vehicles	2.6	2.5, 2.6, 2.7, 2.8, 2.9	m/s^2^
decel	The deceleration ability of vehicles	4.5	4, 4.3, 4.5, 4.8, 5, 5.3, 15.5, 5.8	m/s^2^
emergencyDecel	The maximum deceleration ability of vehicles of this type in case of emergency, >= decel	-	decel + 0, decel + 1, decel + 2, decel + 3	m/s^2^
sigma	The driver imperfection (0 denotes perfect driving) [0, 1]	0.5	0, 0.25, 0.5, 0.75, 1	-
tau	The driver’s desired (minimum) time headway. Exact interpretation varies by model. For the default model, Krauss, this is based on the net space between leader’s back and the follower’s front.	-	0.25, 0.5, 0.75, 0.9, 1, 1.25	s
security	desire for security	-	1, 2, 3, 4, 5	-
estimation	accuracy of situation estimation	-	1, 2, 3, 4, 5	-

**Table 5 sensors-22-09801-t005:** Results of working day data set.

Car-Following Model	Configurations	Total Number of Simulations	Studied Number of Simulations	Minimal Discrepancy %	Deviation in Minimal Discrepancy Routing Algorithm Data Set %	Minimal Discrepancy Routing Algorithm
Krauss	All	450	450	21.598	0	Contraction Hierarchies
Wagner	All	450	450	21.599	0	Contraction Hierarchies
Modified Krauss	Dijkstra only	225,000	828	22.824	0	Dijkstra
Wiedemann	Dijkstra only	16,200	825	22.824	0	Dijkstra
Modified Krauss	Contraction Hierarchies only	225,000	368	21.598	0	Contraction Hierarchies
Krauss	simulation step = 0.01, Contraction Hierarchies only	450	30	32.700	0	Contraction Hierarchies
Krauss	simulation step = 0.001, Contraction Hierarchies only	450	4	30.454	0.932	Contraction Hierarchies
Wagner	simulation step = 0.001	450	2	27.11	4.320	Dijkstra
Krauss	routing step 50 + additional data for 19.02–19.03	450	90	24.941	0	Contraction Hierarchies
Wagner	routing step 50 + additional data for 19.02–19.03	450	90	24.941	0	Contraction Hierarchies

**Table 6 sensors-22-09801-t006:** Results of holiday data set.

Car-Following Model	Configurations	Total Number of Simulations	Studied Number of Simulations	Minimal Discrepancy %	Deviation in Minimal Discrepancy Routing Algorithm Data Set %	Minimal Discrepancy Routing Algorithm
Krauss	All	450	450	46.990	0	Dijkstra
Wagner	All	450	450	46.776	0	Dijkstra
Modified Krauss	Dijkstra only	225,000	1413	46.776	0	Dijkstra
Wiedemann	Dijkstra only	16,200	1409	46.776	0	Dijkstra
Modified Krauss	Additional set	225,000	500	46.776	0	Dijkstra
Krauss	routing step 50 + additional data for 19.02–19.03	450	90	22.649	0	Contraction Hierarchies
Wagner	routing step 50 + additional data for 19.02–19.03	450	90	22.649	0	Contraction Hierarchies

## Data Availability

The sequential methodology implementation is available on: https://github.com/tmecheva/SumoTD (accessed on 10 December 2022). The parallel methodology implementation is available on: https://github.com/dinozavyr/TraficSim (accessed on 10 December 2022). The raw traffic data reports are provided with the consent of the Plovdiv Municipality (available on: https://github.com/tmecheva/SumoTD/tree/main/Reports (accessed on 10 December 2022)).

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
