# Peer review of "Modeling Driver Behavior in Road Traffic Simulation"

_sensors, 2022, doi:10.3390/s22249801_

Round 1

Reviewer 1 Report

This paper is well structured. The reviewers have a few concerns and comments.

1The driver's behavior is different under different road conditions. This paper only distinguishes the modeling results for holidays and weekdays. The research subjects need to be more detailed to match the theme.

2Main contributions in the introduction cannot express the research innovation clearly, more research details and significance are hoped to be added.

3There are a lot of data in the article, data sources, data characteristics and experimental datum processing are not clearly expressed.

4The description of the method in Figure 1 is too simple, please expand the detailed description.

5This paper selects four car-following models and three rooting algorithmsplease explain the reasons for the theories you use in the paper.

6It is recommended that the results be presented in graphical form followed by descriptive analysis.

7Table 3 has various parameter settings, to explain the basis of parameter settings.

8The discussion section of this article is too simple, and a deeper analysis is recommended

9In working daysis the car-follow model also different for the morning and evening peak data and how to choose the corresponding model in this paper?

Reviewer 2 Report

The authors attempted to propose a methodology for establishing the parameters of the drivers’ behavior model. The work needs major revision. My comments are as follows;

==== INTRODUCTION ==== 

The authors should add more references in the introduction to support the claims.

The authors need to better explain the context of this research, including why the research problem is important.

==== RELATED WORK ==== 

The related work section is not well organized. Authors must try to categorize the papers and present them in a logical way.

The authors should explain clearly what  the differences are between the prior work and the solution presented in this paper.

The related work section is too short.

The authors should add a table that compares the key characteristics of prior work to highlight their differences and limitations. The authors may also consider adding a line in the table to describe the proposed solution.

==== PROBLEM DEFINITION ==== 

The authors should add an example to illustrate the problem definition. 

==== EXPERIMENTS ==== 

The experiments should be updated to include some comparison with newer studies. 

A statistical analysis should be carried out to demonstrate that the experimental results are significant. 

There is not enough discussion of the experimental results. 

Some experiment(s) should be added to show that the proposed solution can be used in real applications.

==== REPRODUCIBILITY ==== 

To ensure the reproducibility of the results, the code of the proposed solution should be made public on a website. the links provided are broken, I checked two of them and they were broken

Round 2

Reviewer 1 Report

1In the abstract, the research implications need to be concise to focus. Instead, the methods and findings should be more specifically described, including some experimental results.

2The discussion section of this article is not detailed enough, and more specific discussion and analysis are recommended.

3Whether the results of this paper are generalizable and suitable as a basis for the construction of other models.

4This paper proposes a modeling driver behavior in road traffic simulationCan the modeling in road traffic simulation be more detailed not only on weekdays or weekends.

Reviewer 2 Report

Authors have addressed my comments